# Distinct Features Based on Partitioning of the Endophytic Fungi of Cereals and Other Grasses

Xiang Sun,[a] Or Sharon,[b] Amir Sharon[b]

aSchool of Life Sciences, Hebei University, Baoding, Hebei, China
bSchool of Plant Sciences and Food Security, Tel Aviv University, Tel Aviv, Israel

**ABSTRACT** Endophytic fungi form a significant part of the plant mycobiome. Defining core members is crucial to understanding the assembly mechanism of fungal endophytic communities (FECs) and identifying functionally important community members. We conducted a meta-analysis of FECs in stems of wheat and five wild cereal species and generated a landscape of the fungal endophytic assemblages in this group of plants. The analysis revealed that several Ascomycota members and basidiomycetous yeasts formed an important compartment of the FECs in these plants. We observed a weak spatial autocorrelation at the regional scale and high intrahost variations in the FECs, suggesting a space-related heterogeneity. Accordingly, we propose that the heterogeneity among subcommunities should be a criterion to define the core endophytic members. Analysis of the subcommunities and meta-communities showed that the core and noncore members had distinct roles in various assembly processes, such as stochasticity, universal dynamics, and network characteristics, within each host. The distinct features identified between the community partitions of endophytes aid in understanding the principles that govern the assembly and function of natural communities. These findings can assist in designing synthetic microbiomes.

**IMPORTANCE** This study proposes a novel method for diagnosing core microbiotas based on prevalence of community members in a meta-community, which could be determined and supported statistically. Using this approach, the study found stratification in community assembly processes within fungal endophyte communities (FECs) in the stems of wheat and cereal-related wild species. The core and noncore partitions of the FECs exhibited certain degrees of determinism from different aspects. Further analysis revealed abundant and consistent interactions between rare taxa, which might contribute to the determinism process in noncore partitions. Despite minor differences in FEC compositions, wheat FECs showed distinct patterns in community assembly processes compared to wild species, suggesting the effects of domestication on FECs. Overall, our study provided a new approach for identifying core microbiota and provides insights into the community assembly processes within FECs in wheat and related wild species.

**KEYWORDS** community assembly, core microbiome, endophytic fungi, wheat, wild cereals

Microbial communities are commonly an amalgam of two main subpopulations: one composed of relatively stable and highly abundant taxa, generally referred to as a core population, and a mixture of sporadic taxa with variable abundance. Turnbaugh et al. (1, 2) first proposed the core microbiome concept as "all taxa that are common to the microbiomes in all or the vast majority of habitats," and it has evolved since then. However, Shade and Handelsman (3) suggested that in addition to the shared members or genes among communities, the core microbiome concept should include information about the interaction between the members within the community. Following this, Agler

Address correspondence to Xiang Sun, sunx@hbu.edu.cn.

The authors declare no conflict of interest.

et al. (4) showed how specific taxa shape the microbial community structure via critical interactions with other members of the community. Toju et al. (5) emphasized the importance of interactions among community members and defined core microbiomes as sets of microorganisms that interact to optimize the community functions, even when the core species do not directly influence the host. Delgado-Baquerizo et al. (6) and Shade and Stopnisek (7) defined core microbiome based on abundance-occupancy distributions that include highly abundant and ubiquitous taxa. The core microbiome has been defined in different ways; therefore, to simplify, Shade and Stopnisek (7) and Neu et al. (8) suggested that core microbiome includes the set of microbial taxa, community features, and the associated genomic or functional attributes of the specific host or environment. Recently, Berg et al. (9) reviewed the core microbiome concept and proposed the term "core microbiota", which refers to a "suite of members shared among microbial consortia from similar habitats." Therefore, the core microbiome is a paradigm in microbiome research rather than a clearly delimited research object.

An increasing researches number of studies on plant symbiotic microbes have incorporated the identification of core microbiome, as core members of microbial communities are expected to play pivotal roles in organizing the dynamics of resident microbiomes (5). Chen et al. (10) investigated the endophytes and epiphytes on seeds of the medicinal plant Salvia miltiorrhiza and found that Dothideomycetes taxa overwhelmed the core mycobiome in the study. Thomas et al. (11) found consistent core mycobiome coexisted with the host in wood endophytes, while the core fungi of leaf endophytes were more dynamic, changing across the topography and distance between sampling sites. Grady et al. (12) indicated that switchgrass and miscanthus shared similar membership and dynamics of the phyllosphere core microbiota. Liu et al. (13) reported that the core endophytes drove seasonal community succession of endophytic communities in vineyards. However, the paradigm "core microbiome" is challenged in a plant symbiotic plant microbiomes, because the heterogeneity among the geographic locations or under different circumstances would produce different cores among subcommunities within a common host. Moreover, there are several problems issues related to core microbiome in current researches referring to core microbiome: 1) most studies use empirical criteria would be used to determine core sets, which is arbitrary and not tested with statistical approach; 2) arbitrary criteria might decrease the resolution of research results and lead to loss of lose information; and 3) core conceptualized with abundance data would tend to be biased by epidemic distribution pattern. Therefore, novel definition and detection strategy for core members of microbial communities is needed, to improve our understanding the structure tool microbial communities.

Usually, the taxa that are not included or considered core microbiome are highly diverse with low frequencies and variable abundance. The relatively stable core component and the more variable fraction form the total ecosystem biodiversity, including diversity in species, genetics, and ecological interactions (14, 15). The noncore taxa possess community assembly properties different from those of the core taxa (16). Noncore taxa vary in abundance (17), contributing greatly to community diversity and dynamics. Jousset et al. (18) suggested that the less abundant taxa play an important role in biogeochemical cycles and maybe the overlooked keystone species driving microbiome functions. This component is also highly vulnerable to changes in environmental and other factors (19, 20).

Determining microbial community structure is critical for inferring the functional importance of a certain taxon or taxa and defining the core microbiota. Statistical methods with ecological models have been used to extract information from microbial communities, such as community assembly mechanisms (21, 22) and intracommunity interactions (4, 5). The assembly reflects the influence of selection, dispersal, and ecological drift during community construction (23), which greatly interests microbial ecology studies (24). Stochastic and deterministic processes coexist during the assembly of

plant microbial communities and shift as the plant develops and as environmental conditions change (22, 25–28).

Typically, network analysis is used to infer interrelationships and identify key taxa that determine community features (4, 29, 30). For example, the co-occurrence network has revealed heterogeneity of soil microbial community at the microcosm and continental scales (31, 32), identified antagonistic counterparts among the pathogen and biocontrol agents (33), and uncovered host physical or pathological condition-dependent community changes (34, 35). In particular, a co-occurrence network has been used to deduce the intracommunity interactions among members or distribution patterns at a macro geographic scale (36–38). However, the results of co-occurrence analysis should be interpreted meticulously because of the limitations in statistical models, the influence of broad geographical scales, and the complexity of the natural data (39). Considering the "commonness" emphasized in the core microbiome concept, the co-occurrence associations shared among subcommunities should be addressed meticulously.

Endophytes are symbiotic microbes living within healthy plants (40). Over the past years, numerous endophytes have been sourced from domesticated and wild plants (41–44). However, additional research is needed to advance our understanding of the processes that affect the assembly of and shape the endophytic communities. Here, we incorporated data from *Triticum aestivum* (bread wheat) in agricultural fields and five wild grass species from natural habitats within the boundaries of Israel (26, 45) and produced a high-resolution landscape of stem fungal endophyte communities (FECs). Based on our findings, we propose a modified definition and diagnostic criteria for core endophyte mycobiomes in the present scenario. We investigated the distinct features and importance of core and noncore taxa in community assembly and identified differences in the features of FECs between core and noncore partitions in wheat and wild species.

## RESULTS

**General description of the actual FECs.** After quality control, the internal transcribed spacer 1 (ITS1) amplicon sequencing data set generated 9,516,076 reads, including 1,336 taxa from 1,032 samples. Further filtering and Hellinger transformation resulted in a final data set of 1,312 taxa from 916 samples, which were used for subsequent analysis.

Our analysis revealed that 99.42% of the community was composed of fungi from the 12 predominant classes or higher taxonomical hierarchies. The top 20 species and operational taxonomic units (OTUs) accounted for 63.76% of the community, with *Alternaria infectoria* being the most abundant species. Fungi of the class Dothideomycetes dominated all plant populations (relative abundance [RA] = 47.64%) and were more abundant in the five wild plant species (RA was 45.54% in *Aegilops peregrina*, 56.07% in *Aegilops sharonensis*, 52.25% in *Avena sterilis*, 49.22% in *Hordeum spontaneum*, and 45.60% in *Triticum dicoccoides*) than in the cultivated wheat (RA = 40.73%) (Fig. 1a). The second most abundant class was Tremellomycetes (Basidiomycota), which accounted for 22.91% of all fungal classes in the pooled data sets. We also found that Saccharomycetes were highly abundant in wheat (Fig. 1a), of which *Candida sake*, with a RA of 14.15%, was the predominant one (Fig. 1b). Taxa belonging to the class Sordariomycetes were highly abundant in wheat and the wheat ancestors *T. dicoccoides* and *Aegilops sharonensis* but not in the other plant species, highlighting the potential association of these taxonomic groups with wheat and its ancestors (tribe *Triticeae*) (Fig. 1b). The FECs differed in composition among the wheat and its wild relatives and showed high intrahost variations (Fig. 1c). However, no notable latitude-correlated pattern in community composition (Fig. 1d) and geographical distance-related effect on wheat FECs were observed at the regional scale (Fig. 1e).

**Core taxa.** Core taxa of the real and simulated communities were screened using thresholds from 99% to 0%. The number of core taxa in all six species demonstrated a sharp L curve in the real and simulated communities with an increase in the threshold, while the abundance of core sets decreased evenly (Fig. 2). In this analysis, several threshold intervals yielded the same core sets in real and simulated data sets ($J = 1$) for each

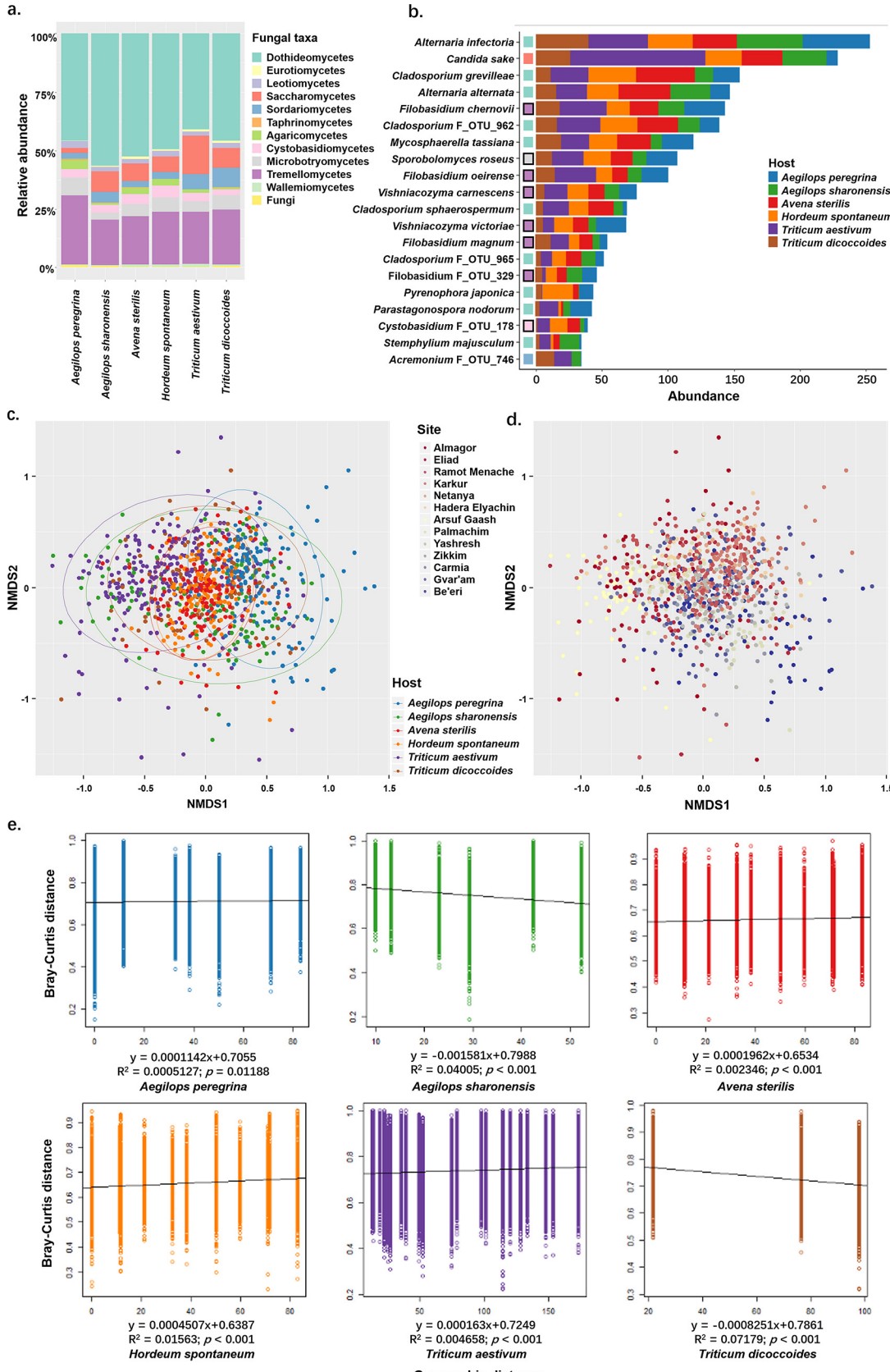

**FIG 1** Community composition of the stem endophytes of wheat and wild cereals. (a) Relative abundances of the 12 most abundant fungal classes or higher taxa in the meta-community. (b) Hellinger-transformed abundance (*x* axis) of the 20 predominant

host. The series of core sets were named tier 1, tier 2, tier 3, and so on, from high to low thresholds (Fig. 2). Table 1 shows the list of core taxa at different tiers in the six hosts. *Alternaria* spp., *Cladosporium* spp., and *C. sake* were frequently identified as core members in all hosts. Meanwhile, *Ae. peregrina* had the highest number of core taxa among the six species, while wheat had the lowest. The high-tier core sets usually showed low abundance. The tier 1 core set in *Ae. peregrina* accommodated a single taxon, *Cladosporium grevilleae*, that accounted for 3.85% of the overall abundance in the meta-communities of the six species (Fig. 2). We chose the threshold intervals that yielded core sets with abundances close to 50% to define the core and noncore sets in subsequent analysis (the tiers with black frames in Fig. 2). For example, the tier 2 core set, which included 17 taxa and accounted for 55.38% of the abundance, was chosen in *Ae. peregrina*.

**NCM fit and universal dynamics.** The FECs of the six host plants showed a good fit to the neutral community model (NCM) and relatively low $Nm$ values (Fig. 3a). Most fungal taxa throughout all communities were distributed between the confidence intervals (black dots in Fig. 3a). Nevertheless, when communities were separated into core and noncore partitions (above and below horizontal dashed lines in Fig. 3a), the ratios between taxa that fit or did not fit the NCM were different. The taxa with higher prevalence (Fig. 3, turquoise dots) or abundance (Fig. 3, red dots) in the core partitions were detected at higher proportions than those in the noncore partitions. These observations suggested that the core sets were more likely composed of taxa that deviated from the NCM prediction and showed a deterministic pattern to some degree; the opposite was true for the noncore sets. The low Nm values revealed in our research indicated limited migration amongst the FECs of all hosts. This, which could lead led to more isolated and differentiated local communities, which is consistent with the high intra-host variation shown in Fig. 2c.

When the threshold of the core set shifted from 0% to 99%, the proportion occupied by taxa that deviated from NCM prediction increased as criteria became rigorous in all hosts except *T. dicoccoides* (stacked area charts in Fig. 3a). Meanwhile, the noncore sets were consistently overwhelmed with neutral taxa. Here, *Ae. sharonensis* and wheat possessed more highly abundant core taxa as thresholds increased, suggesting that the extremely abundant endemic taxa largely influence their core endophytes at certain locations or in specific individuals. In particular, *C. sake*, the tier 1 core taxon in wheat, was more abundant than the prediction (Table 1). In contrast, *Ae. peregrina*, *A. sterilis*, and *H. spontaneum* possessed more prevalent taxa in their core sets with increase in the thresholds, suggesting that their core taxa include a large portion of symbiotic partners. Nevertheless, the proportion of neutral taxa within the core taxa of *T. dicoccoides* remained high even at high thresholds, suggesting the role of a stochastic process in regulating the assembly of the *T. dicoccoides* stem FEC.

The existence for universal dynamics would be suggested with the trends of upward DOCs turning to downwardly at right of change points. Dissimilarity overlap curve (DOC) analysis of meta-communities showed inflected curves in four hosts and declined after the change point (vertical black line in Fig. 4c); these observations suggested universal dynamics of the FECs in these plants (see Fig. S1 in the supplemental material). The upward DOC generated by FECs from wheat and *A. sterilis* showed no universal dynamics. When the DOCs of core and noncore partitions for FECs in each host were separately calculated, we obtained different results from the two partitions. We found upward or roughly flat curves in all core partitions and no meaningful difference between the real data and null models (Fig. 3b), suggesting no universal dynamics in the core partition of the communities. Meanwhile, for the noncore partitions, the upward trends of DOCs were interrupted

**FIG 1** Legend (Continued)
fungal species or OTUs in the meta-community. Colored squares to the right of species or OTU names indicate the fungal taxa; the color key is shown in panel a. Squares with black frames indicate the basidiomycetous species with a yeast lifestyle. (c) NMDS plot of community composition based on Bray-Curtis dissimilarity (stress = 0.2577) labeled according to host. (d) NMDS plot of community composition based on Bray-Curtis dissimilarity labeled according to site; sites are arranged from north to south. (e) Linear modeling between community dissimilarity and geographic distances. The low $R^2$ values and slopes close to zero indicate that geographic distance had a negligible effect on community composition at the current scale.

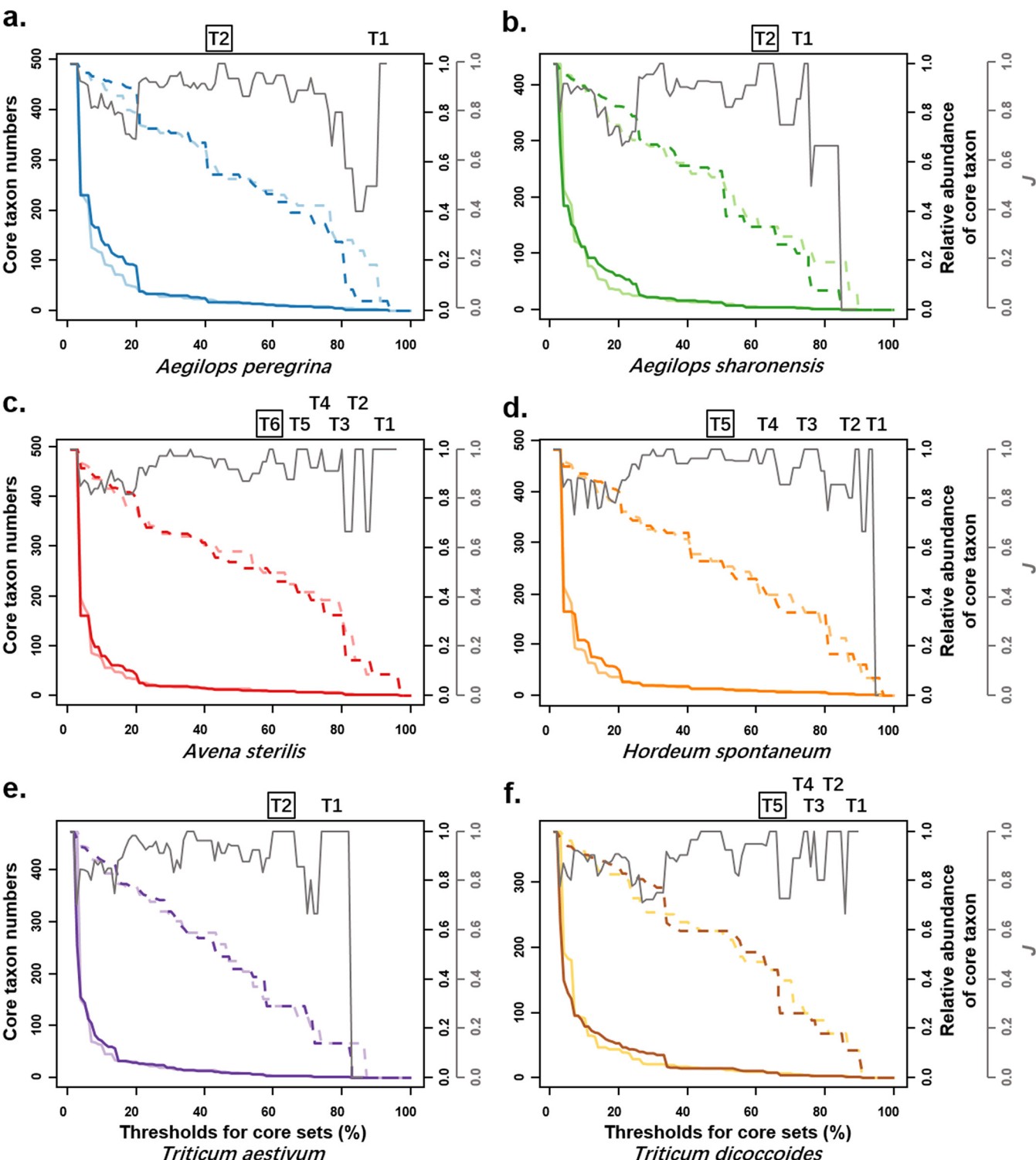

**FIG 2** Taxon number and abundance in the core sets of real and simulated communities. (a)–(f) indicated results from all six hosts. The *x* axes indicate thresholds ranging from 0% to 99%. Colored solid lines show numbers of core taxa, and dashed lines represent the sum of the relative abundance of core taxa. Dark colors show values calculated with real communities, and light colors show values calculated with simulated communities. Gray solid lines indicate *J* between core sets identified with real and simulated communities at certain thresholds. At a *J* value of 1, actual and simulated communities yielded the same sets of core taxa; these are labeled as tier 1 (T1), etc. Black frames around tier numbers indicate core sets used in the subsequent analysis as core endophytes of a specific plant species.

in the five wild species. Clear downward trends could be observed in *Ae. peregrina* and *H. spontaneum*, and DOCs seemed to reach slightly downward zone or plateau in *Ae. sharonensis*, *A. sterilis*, and *T. dicoccoides*. The change points of DOCs in five wild species were also more leftward on the *x* axis than those in null models. Moreover, in all five wild

**TABLE 1** Different tiers of core community in all six host plants

| Organism | Tier in[a]: | | | | | |
|---|---|---|---|---|---|---|
| | *Aegilops peregrina* | *Aegilops sharonensis* | *Avena sterilis* | *Hordeum spontaneum* | *Triticum aestivum* | *Triticum dicoccoides* |
| Ascomycota | | | | | | |
| Dothideomycetes | | | | | | |
| Capnodiales | | | | | | |
| Cladosporiaceae | | | | | | |
| *Cladosporium grevilleae* | 1 (F) | 1 (F) | 1 (F) | 1 (F) | 2 (F) | 4 (F) |
| *Cladosporium sphaerospermum* | | | 5 (F) | 4 (F) | | |
| *Cladosporium* F_OTU_962 | 2 (F) | 2 (N) | 2 (F) | 2 (F) | 2 (N) | 5 (N) |
| *Cladosporium* F_OTU_965 | 2 (F) | | | | | |
| Mycosphaerellaceae | | | | | | |
| *Mycosphaerella tassiana* | 2 (F) | | 3 (F) | 3 (F) | | 3 (N) |
| Pleosporales | | | | | | |
| Phaeosphaeriaceae | | | | | | |
| *Phaeosphaeria* F_OTU_624 | 2 (N) | | | | | |
| Pleosporaceae | | | | | | |
| *Alternaria alternata* | 2 (F) | 2 (N) | 3 (N) | 3 (N) | | 5 (N) |
| *Alternaria infectoria* | 2 (A) | 1 (N) | 3 (N) | 3 (N) | 2 (N) | 1 (N) |
| Saccharomycetes | | | | | | |
| Saccharomycetales | | | | | | |
| *Incertae sedis* | | | | | | |
| *Candida sake* | 2 (F) | 1 (N) | 4 (N) | 3 (N) | 1 (A) | 2 (N) |
| Basidiomycota | | | | | | |
| Cystobasidiomycetes | | | | | | |
| *Incertae sedis* | | | | | | |
| Symmetrosporaceae | | | | | | |
| *Symmetrospora coprosmae* | 2 (F) | | | | | |
| Microbotryomycetes | | | | | | |
| Sporidiobolales | | | | | | |
| Sporidiobolaceae | | | | | | |
| *Sporobolomyces roseus* | 2 (N) | | 6 (N) | 4 (N) | | |
| Tremellomycetes | | | | | | |
| Filobasidiales | | | | | | |
| Filobasidiaceae | | | | | | |
| *Filobasidium chernovii* | 2 (N) | | 6 (N) | 5 (N) | | 5 (A) |
| *Filobasidium oeirense* | 2 (N) | | | | | 5 (N) |
| *Filobasidium* F_OTU_329 | 2 (N) | | | | | |
| Tremellales | | | | | | |
| Bulleribasidiaceae | | | | | | |
| *Dioszegia buhagiarii* | 2 (F) | | | | | |
| *Vishniacozyma carnescens* | 2 (N) | | | 5 (N) | | |
| *Vishniacozyma dimennae* | 2 (N) | | | | | |
| *Vishniacozyma victoriae* | 2 (N) | | | | | |

[a]Lower tiers included members in higher tiers (e.g., tier 3 core members of *A. sterilis* include all taxa labeled as tier 1, 2, and 3, while tier 4 core members include all taxa labeled as tier 1, 2, 3, and 4). Letters in parentheses indicate prediction in NCM of the taxon: F, more frequent than prediction (above the confidence intervals in Fig. 3); A, more abundant (lower in Fig. 3); N, neutral and fitting the prediction (between intervals in Fig. 3).

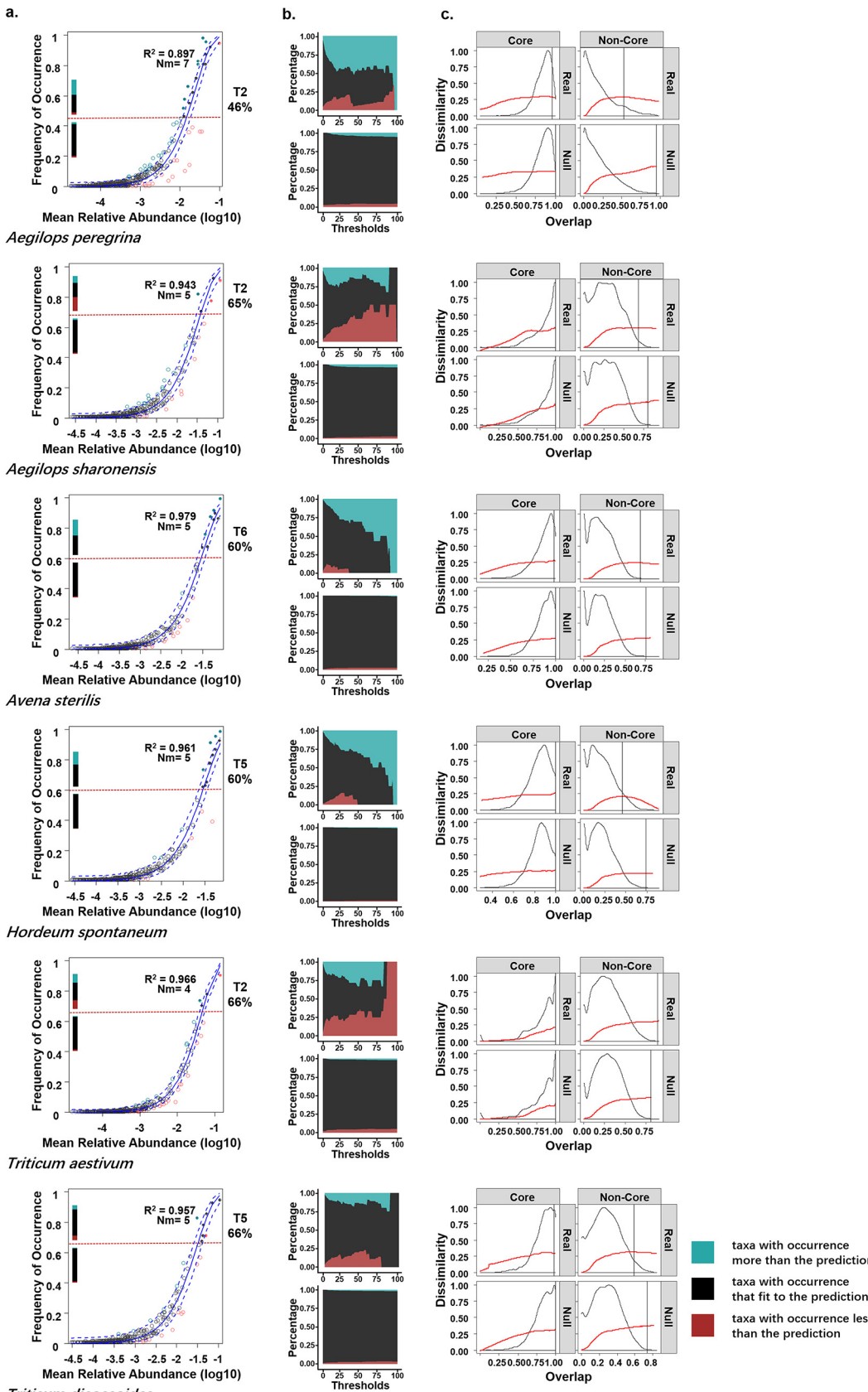

**FIG 3** NCM fit and DOC of core and noncore taxa. (a) NCM fit of each taxon. The plots show the NCM fit of a taxon in each host. Solid blue lines indicate the best fit to the neutral model, and dashed blue lines represent 95% confidence

species, the curves showed a steep downward trend passing through the change point in the actual data compared to the null models. However, DOC of *T. aestivum* non-core partition kept upward, and thus no change point existed on curve while the vertical line was located at the rightmost of the x-axis. Collectively, the DOC analysis revealed notable universal dynamics for the communities of the noncore partitions from all five wild plant species but not from cultivated *T. aestivum*.

**Shared co-occurrence associations within the FECs.** We further performed co-occurrence network analysis to dissect the features of the communities in different partitions, particularly the noncore sets. The dissimilarity analysis indicated that networks generated from real and simulated data sets were similar and comparable, as most of the real networks were distributed together with simulated networks within a limited range despite a few outliers in nonmetric multidimensional scaling (NMDS) plots (Fig. S2). We then investigated the common co-occurrence associations shared among the subcommunities based on the real and simulated data. A large proportion of the associations from the 100 simulated networks in each host was present in only one or a few subcommunities (Fig. 4a). We selected all the associations present in more than 20 subcommunities and combined them into a single network graph (Fig. 4b). Surprisingly, the most frequent associations (edges in a network graph) were generally between the taxa with low abundance, while the abundant or core taxa formed hardly any co-occurrence associations (Fig. 4b). The combined networks were formed by the scattered modules rather than the connected vertices (Fig. 4b), which suggested a notable pattern in community assembly, where the stable associations were formed by the co-occurring rare taxa of the small groups. The combined network from wheat was different from those of the wild plant species. The wheat endophytes showed an extraordinarily sparse network composed of only 12 fungal taxa with relatively low frequencies and abundances. Moreover, most of the vertices in the combined co-occurrence networks of the six hosts represented neutral taxa that fit the NCM (Fig. 4b, black), with a small proportion of less frequent taxa and hardly any frequent taxa (turquoise) compared with the NCM prediction. Among the networks generated with the real or simulated communities, the taxa (vertices in networks) with high degrees (numbers of edges) and centralities were predominantly less frequent than the prevalent taxa for all six plant species (Fig. S3).

## DISCUSSION

**Defining the core endophytes in wheat and its wild relatives.** Microbial communities display spatial variations in composition, as geographic distance leads to heterogeneous environments and species pools and results in differentiated selection (46). Thus, the spatial factor should also be considered when the core microbiota is being defined. Studies have shown dissimilar FECs of wheat at the fungal species level compared with other plant species and across different geographic locations (22, 47–49). Nevertheless, studies on cereal endophytes from a wide range of geographic locations and conditions showed a relatively conserved spatial distribution pattern of fungal endophytes at the class level at regional (45, 50) and continental (51, 52) scales (Table S3). Therefore, we conclude that the cereal and related plants probably possess core sets of FECs relatively unaffected by spatial heterogeneity.

**FIG 3** Legend (Continued)

intervals with 1,000 bootstrap replicates. Black dots between dashed blue lines indicate taxa that fit the neutral model with 95% confidence. Taxa that deviate from the model and occur more or less frequently than the predictions are shown in turquoise and red, respectively. $R^2$ indicates the fit to the neutral model. $Nm$ is meta-community size times migration rate. Horizontal dashed lines indicate thresholds for certain tiers of the core set determined in Fig. 2, which divide the plots into core taxa (top; solid circles) and noncore taxa (bottom; open circles). Stacked bars above and below the dashed line indicate taxon proportions for core and noncore partitions; the colors used for the taxa are the same as for the plots. (b) Taxon proportions for core and non-core partitions at different thresholds. Proportions of taxa that fit or deviate from the NCM in the core and noncore sets with an increase in the thresholds of core from 0% to 99% (x axis). (c) DOC of meta-communities in the core and noncore partitions of each host. DOC (red) were calculated for core and noncore sets to detect universality in the dynamics of endophyte communities in different partitions. The overlap distributions of the real and randomized between-subject sample pairs are shown as black curves. The vertical black lines represent the change points.

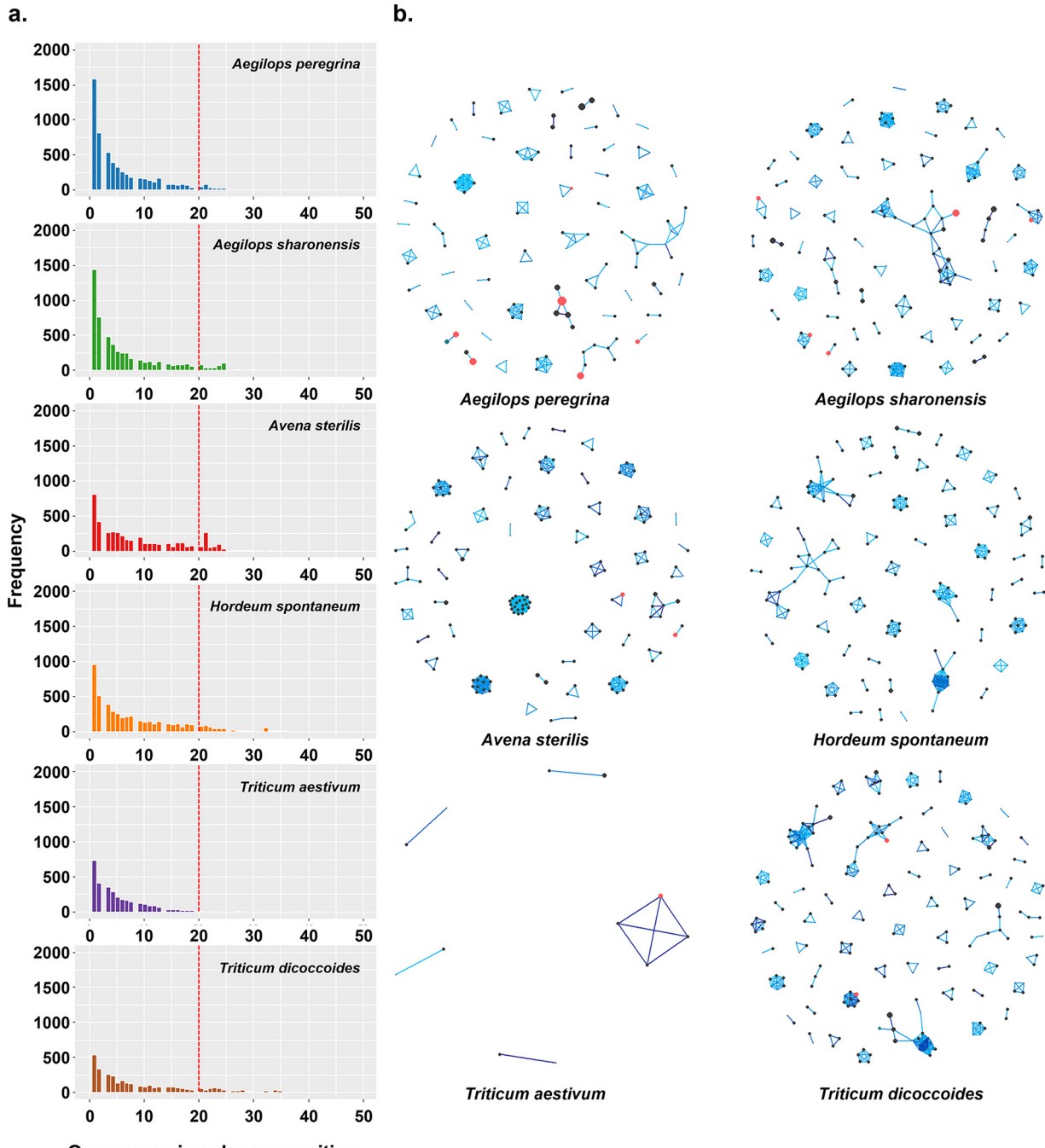

**FIG 4** Combined co-occurrence networks of the six hosts based on the simulated data. One hundred co-occurrence networks were generated from simulated communities with a threshold |*rho*| of >0.8 and a *P* value of <0.001; then, the shared edges (associations) presented in multiple simulated networks were combined to generate a single network graph (see the supplemental material for details). (a) Histograms showing the numbers of all edges present among the 100 networks generated from simulated communities with certain frequencies. Only a few edges occurred more than 20 times in 100 networks (to the right of the red line). (b) Combined co-occurrence networks of edges that occurred more than 20 times among 100 simulated subcommunities in each of the six hosts. Vertex size indicates the relative abundance of the taxon; vertex color indicates whether the taxon fits the neutral model (black, fit; turquoise and red, not a fit [as in Fig. 3]). The color of edges indicates the frequency of the edge in 100 stimulated networks from low (light) to high (dark). The taxonomical identities of nodes in the network graphs are listed in Table S2.

The core taxa of the six plants surveyed in this study included common ubiquitous opportunists, such as *Alternaria* spp. and *Cladosporium* spp., which may switch among endophytic, saprophytic, and pathogenic lifestyles depending on conditions (53, 54). One notable taxon is *A. infectoria*, the most abundant fungal endophyte in the cereal

FECs (45, 55). Although *A. infectoria* is known as a wheat pathogen (56–60), the endophytic strains is found in the stems and seeds of wheat and its wild relatives (55, 61, 62), suggesting that it is an integral part of the FECs of these plants. Meanwhile, *C. sake*, the most abundant endophyte identified in wheat, has demonstrated biocontrol properties in other plant species (63, 64) but has never been isolated from wheat or identified as a wheat endophyte using traditional techniques. We attempted to compare the biocontrol properties of the wheat-borne *C. sake* and the isolates from other species; however, we could not isolate *C. sake* from wheat plants despite intensive efforts (data not shown). The study identified another prevalent taxon, *Mycosphaerella tassiana*, defined as a tier 2 core taxon in *Ae. peregrina* and a tier 3 core taxon in the other three hosts. The genus *Mycosphaerella* includes numerous well-known plant pathogens (65). However, researchers have recovered *M. tassiana* from Cruciferae plants and palms (66, 67) and reported it as a wheat endophyte in conventionally tilled agricultural fields (68). Compared to the earlier reports, our current study revealed new ecological niches of fungal taxa as core endophytes of wheat and cereal-related plants. The work also suggested the taxonomical uniqueness of endophytes on the surveyed plants of this region; thus, stronger heterogeneity in endophyte taxonomical composition could be inferred at a larger spatial scale.

**Basidiomycetous members among the endophytes of wheat and its wild relatives.** Numerous taxa of the phylum Basidiomycota dominated the endophyte assemblages of all six plants and were diagnosed as core taxa in *Ae. peregrina*, *A. sterilis*, *H. spontaneum*, and *T. dicoccoides* (Table 1, Fig. 1). Other studies demonstrated similar results in wheat endophytes at the continental scale (47, 51, 69) (Table S3). Intriguingly, most of the Basidiomycota taxa belonged to the classes Tremellomycetes, Microbotryomycetes, and Cystobasidiomycetes, which have a yeast phase. We refer to these species as endophytic basidiomycetous yeasts (EBYs). EBYs can be considered "dark matter" in wheat endophyte assemblage (70, 71). They represent up to 50% of the wheat FECs in abundance (47, 51, 69), and a few seem indispensable components of the wheat FECs. However, only a few EBYs have been successfully recovered from plants (55, 72, 73). Thus novel techniques to culture EBY *in vitro* are essential for future studies on wheat FECs.

EBY assemblages are influenced by external factors such as provincial circumstances, local microbial species, and wheat cultivars (50, 74). In addition, EBYs are found to be colonize stems, leaves, and immature spikes but not in roots (48, 49) and tend to decline during kernel development in wheats (52, 75). EBYs are promising plant-beneficial microbial candidates as biocontrol agents and could be functional modules in wheat mycobiota synthetic communities (44, 51, 76). Present study has identified *Filobasidium* spp., *Dioszegia* spp., *Vishniacozyma* spp. (Tremellomycetes), *Sporobolomyces roseus* (Microbotryomycetes), and *Symmetrospora coprosmae* (Cystobasidiomycetes) as the core endophytes in four of the six surveyed plants. No EBY met our criteria for core taxa in *Ae. sharonensis* and *Triticum aestivum*; however, they still maintained a significant presence in their FECs (Fig. 1a and b). A few EBYs, such as *Filobasidium* spp., *Sporobolomyces roseus*, and *Vishniacozyma victoriae*, have been reported as dominant endophytes in wheat (45, 47–50, 52, 77), and *Filobasidium* spp. were only recently detected as endophytes in wheat in Italy and Israel (55, 57). Their high abundance in the Mediterranean region contrasts with their low levels in wheat FECs from other parts of the world (49, 78, 79), probably due to the wheat center of origin and the specific climate conditions in this region. Thus, our findings emphasize EBYs as the key components in FECs of wheat and cereal-related plants.

**Inner partitioned features of community assembly.** Deterministic and stochastic processes affected the assembly of the natural or artificial FECs. Increasing evidence indicates that the stochastic process has a critical role in microbial community assembly in natural ecosystems (24), and different partitions of the community might assemble either under both or one of these processes (16, 80, 81). In several cases, the assembly of main players in the community followed a deterministic process, while minor taxa followed neutral processes (82, 83). Similarly, we noticed determinism for core partitions and stochasticity for noncore partitions in all host species, except *T. dicoccoides*, based

on the NCM prediction. These observations suggested that deterministic and stochastic effects usually work together within natural communities.

Generally, DOC detects host-independent (universal) dynamics in microbial communities, which can be estimated based on growth rates and intra- or interspecies interactions in dynamic population models (84). The method has been used in studies on the human microbial community (84) and arbuscular mycorrhizal fungi (85). To our surprise, the DOC analysis revealed that noncore sets in the FECs exhibited stronger universal dynamics than core sets. This observation implies determinism rather than stochasticity in noncore sets, which is opposite to the NCM predictions. The deviation from the neutral state in NCM indicated the lack of universal dynamics in core sets. The higher abundance predicted by NCM for most core taxa indicated host-specific colonization patterns, such as plant selection or dispersal limitation (7); it probably caused the lack of universal dynamics in core sets.

In summary, the roles of stochastic and deterministic processes in community assembly were different between the different partitions. Moreover, the various statistical techniques, such as NCM and DOC analysis, highlighted the different effects of the community components on assembly mechanisms. The core sets deviated from the NCM and exhibited deterministic properties, probably due to the hosts' and fungi's inherent traits, such as compatibility and endemic colonization. In contrast, most noncore taxa fit the NCM, while DOC detected universal dynamics for noncore sets, suggesting a certain degree of determinism.

**Differences in the network feature between core and noncore taxa.** The co-occurrence network has been widely used in microbial community analysis to measure hypothetical interactions between community members. To resolve the contradicting results of the NCM and DOC models for determinism versus stochasticity presented in core and noncore sets, we performed a co-occurrence analysis to reveal the interactions among taxa in different partitions. To avoid possible bias by spatial extent (39), we analyzed the co-occurrence network for local (sub)communities and not for the metacommunity. According to Shade and Handelsman (3), core taxa should include members covarying within a community and shared across communities. Previous studies focused on associations shared by co-occurrence networks in wheat and cereals from different sites (45). We included simulated subcommunities in our study to identify overlapping components and evaluate the dynamic features of core or noncore taxa.

Generally, in an ecological network of community, the taxa with a high degree and closeness centrality and the lowest betweenness centrality are considered keystone or hub taxa (86), and the edges with high frequencies are considered core interactions (3, 5). Our results showed that noncore taxa formed more co-occurrence associations and influenced the networks more greatly than core taxa. However, noncore taxa could not establish connections through the network. This structure suggests that the ideal keystone taxa or hub taxa, which construct scaffolds, expand their influence to the entire community or recruit crucial members, as suggested by Toju et al. (5) and Agler et al. (4), might be absent in the FECs of in these plant species. The randomly occurring, neutral noncore taxa seem to interconnect constantly and form small modules of two to five taxa in the networks. Such small modules in networks typically could be recognized as motifs, which is a fundamental unit in a network and an important concept in the emergence of complex system theory (87). In co-occurrence networks of surveyed FECs, the occurrence of such motifs appears stable instead of the occurrence of particular identities of endophytes. Thus, future analysis of the ecological networks of the FECs needs to focus on such self-organized fundamental units.

**Core and noncore taxa show different ecological features.** Our study demonstrated a hierarchical effect of community dynamics within endophytic microbiotas. Studies have shown that different ecological traits influence different community partitions. Petrini (88, 89) concluded that dominant endophytic fungal taxa are more often determined by the host, while the rare taxa usually occur more randomly. Based on our studies of FECs in wheat and the wild cereal relatives, we propose that the core members and noncore members exhibit distinct ecological patterns, maintaining their

existence in a community. However, we still lack sufficient knowledge to understand stratification in FECs. In the open ocean, phytoplankton taxa adopt different strategies of nutrient and light utilization and occupy different ecological niches (90). In intestine microbiota, the metabolic flows among microbes are revealed which formed multiple levels of trophic organizations (91), and the metabolic interactions among microbiota members were exploited to synthesize stable core microbiome (92). Nevertheless, there is no solid evidence whether microscale niche differentiations or metabolic interactions of FEC members exist within plant tissues. The performance of core sets within the FECs in the current study might be attributed to biological traits of fungi and plants, such as compatibility and endemic effects, while noncore sets are assumed to have incidental occurrence and frequent connections. The relatively intensive interactions among noncore taxa revealed by network analysis are consistent with the results of DOCs that displayed universal dynamics. However, the patterns revealed in our study demand an investigation into whether an FEC represents a holistic system, with integrated functions and self-organized infrastructures (e.g., motifs) with possible emergent properties, or a fragmented system composed of weakly associated members and isolated partitions with distinct dynamics.

**Shift in FEC features upon wheat domestication.** Our results highlighted differences in the core sets and community features of the stem FECs from wheat and wild plant species. Compared with wild species, wheat had FECs with fewer core taxa, exhibited almost no consistent intracommunity interactions, and lacked universal dynamics. These differences collectively indicate a reduction in the diversity and a change in the composition of the FECs in the cultivated wheat compared with the wild plants, possibly reflecting the lower genetic diversity of wheat and the conserved farming conditions relative to natural habitats (93). Abdullaeva et al. (94) reported structured, mature seed microbial communities in wild species with more intermicrobe interactions than in domesticated wheat and barley. Domesticated crops are also less dependent on mycorrhiza partners (93). Similarly, our results support the possibility that wheat lost a part of the stable component of fungal symbionts during domestication, possibly leading to communities that lack inner interactions and have reduced universal dynamics. Raaijmakers and Kiers (95) proposed that "wild" microbiotas of crops lost through domestication and industrialization could be reinstated to improve host health. Therefore, it is critical to exploit microbiotas from ancestral or wild species to increase the sustainability of food production in a changing climate.

**Conclusions.** The present study investigated the stem FECs of domesticated wheat (*Triticum aestivum*) and five cereal species based on properly defined core criteria. The study found overlap in the taxonomical composition of the FECs among the six species and locations. However, wheat FECs differed from the wild species' FECs in composition, core taxon sets, universal dynamics, and intracommunity interactions, demonstrating an apparent effect of domestication on the endophytic mycobiota. We also found distinct community assembly features between the core and noncore sets of the FECs. The core members were more determined by host colonization, lacked universal dynamics, and were less connected to other members in the community than the noncore members. We also found that the core taxa deviated from the neutral state predicted by the NCM, while most noncore taxa showed a good NCM fit. Universal dynamics were detected for the communities of the noncore partition, probably due to the extensive co-occurrence associations among these noncore taxa. Our findings provide novel insight into the components and assembly of FECs, which has implications for designing synthetic microbiomes. These principles are possibly also relevant for microbial communities in other types of ecosystems. Therefore, we should dissect the community assembly mechanisms and the effects of different partitions within the microbial community.

## MATERIALS AND METHODS

**Data collection.** Plant sample collection, surface sterilization, DNA extraction, and Illumina sequencing were carried out as reported earlier (26, 45). Samples of *Aegilops peregrina* (goatgrass), *Aegilops sharonensis* (Sharon goatgrass), *Avena sterilis* (wild oat), *Hordeum spontaneum* (wild barley), *Triticum aestivum* (bread wheat), and *Triticum dicoccoides* (wild wheat) were collected in Israel during March and April of 2017 and 2018 when the plats have fully developed spikes and start to flower (Table S1). Healthy plants

of each population (species $\times$ location combinations) were randomly selected in 100 m² (for *Ae. peregrina*, *A. sterilis*, and *H. spontaneum*) or 250 m² (for *Ae. sharonensis*, *T. aestivum*, and *T. dicoccoides*) plots. *T. aestivum* plants were sampled from commercial agricultural fields, while the other five species were sampled from their natural habitats (Table S1).

**Quality control and taxonomy assignment (based on ASVs and OTUs).** The amplicon libraries targeting the internal transcribed spacer (ITS) 1 region were produced using ITS1f (5'-CTTGGTCATTTAGAGGAA GTAA-3') and ITS2r (5'-GCTGCGTTCTTCATCGATGC-3') primers and sequenced with Illumina MiSeq sequencer (Illumina, San Diego, CA). The forward and reverse reads of ITS1 amplicons obtained via Illumina sequencing were demultiplexed, and a quality control check was performed in the QIIME2 platform (version 2018.11). The primers were trimmed from the reads using the Cutadapt tool (96). Subsequently, the DADA2 workflow (97) was used for quality filtering, paired-end merging, and dereplication. This process yielded 3,438 amplicon sequence variants (ASVs), from which the chimeric amplicons were identified using the removeBimeradenovo function and discarded.

The taxonomy was assigned for the fungal ASVs using the naive Bayes approach (minimum, 75 bootstrap calls) following the DADA2 workflow (97) against the UNITE general FASTA release for fungi (version 8.0) (98) and then agglomerated at species level using the tax_glom function in the phyloseq package (97, 99). The nonassigned ASVs were clustered into OTUs based on 97% similarity with the OTU function in the kmer package (100), and a random sequence representing the OTU was conducted to the naive Bayes approach for taxonomy assignment. Details on the procedures are included in the supplemental material.

**Endophytic fungal community description.** The singletons, doubletons, and samples with fewer than 1,000 sequences in each plant species data set were removed, generating 1,312 taxa across 916 samples. The reads of taxa among the samples were then Hellinger transformed with the decostand function in the R vegan package (101). The Hellinger transformed values were treated as abundance in subsequent statistical analysis. Further, the distance matrix of samples was generated following the Bray-Curtis method and ordinated with NMDS with the rarefied data set using the metaMDS function in vegan (101). Spatial autocorrelation of the FECs was fitted to a linear model with the lm function in R (102).

**Definition of a core mycobiota.** We defined a taxon as a core member based on its prevalence within a local community (subcommunity) and how the prevalence of this pattern recurred amid local communities in the meta-community. The prevalence represents is the occurrence ratio of a certain endophytetaxon among plant individuals, which is defined as the numbers of plant individuals colonized with a certain endophytetaxon divided by the number of all plant individuals in the community. First, a taxon of a local community with a frequency of occurrence above a given threshold was defined as a core′ taxon. Then, a taxon determined as a core′ taxon in local communities at more than a given threshold was considered a core taxon in the meta-community (Fig. 5a). For example, if the threshold was set to 50%, a core′ taxon should have a frequency of at least 50% in a local community, and the taxon identified as core′ in more than half of the local communities was defined as a core taxon in the meta-community. The core′ taxa were determined with the formula {$tax_i$ | $occurrence_i/N_{samp} >$ thres} (Equation 1), and the core taxa were determined with the formula {$tax_i$ | $n(tax_i \in core'_j)/N_{comm} >$ thres} (Equation 2), where $i$ is the $i$th taxon, $j$ is the $j$th subcommunity (or local community) in the meta-community, core′ is the expected core member with a frequency above the given threshold in a subcommunity, $tax_i$ is the $i$th taxon in meta-community, $occurrence_i$ is the number of times $tax_i$ occurred in samples in a subcommunity, $N_{samp}$ is the number of samples in a subcommunity, thres is the threshold assigned, core is the core taxon set, $n$ is the cardinal number of ($tax_i \in core'_j$), and $N_{comm}$ is the number of subcommunities in the meta-community.

**Determination of core fungal endophytes.** In this study, the meta-community had 111 to 240 individual samples from each host, dispersed between three to seven sites due to natural distribution and other practical reasons (Table S1). Considering the unevenness of samples, we identified the core taxa with real, local community data and tested their robustness following the bootstrapping strategy. For each host, we identified core taxa of meta-community with a certain threshold. Then, we randomly selected 30 samples (subsampling repeated 100 times) to generate 100 simulated communities from the meta-community of each host and then identified the core taxa with the same threshold for the simulated subcommunities. The dissimilarity between the core member sets calculated with real and simulated communities at a certain threshold was represented with the Jaccard index ($J$). The core set was determined when the real and simulated communities yielded the same core members ($J = 1$). In practice, at thresholds ranging from 0% to 99%, several intervals would yield $J$ values of 1; the core sets determined with thresholds in such intervals were named tier 1, 2, 3...$n$ cores. The complement partitions in the meta-community were named tier 1, 2, 3...$n$ noncores (Fig. 1b, c).

**Community assembly differences between core and noncore communities.** The NCM was used to evaluate the stochasticity in the assembly of communities (103). A Hellinger-transformed data set was multiplied by 1,000 and rounded off, as NCM calculation deals with integer data. The NCM calculation and result demonstration were performed using the R scripts provided by Burns et al. (104) and Chen et al. (105). Then, the 95% confidence intervals around all fitting statistics were calculated by bootstrapping with 1,000 replicates. The numbers of taxa that fit or deviated from the prediction model in core partition and noncore partition in a meta-community were demonstrated as percent stacked area charts. DOC was used to characterize the differences in dynamics between core and noncore communities (84), which was calculated for core and noncore partitions using the DOC package in R (106).

**Co-occurrence network and shared association.** Co-occurrence association among the endophyte species or OTUs were evaluated, and Spearman's *rho* was calculated using the cor.test function in the stats package (102). The associations (edges in the network graph) were calculated for each subcommunity, and those with |*rho*| values of >0.8 and *P* values of <0.001 were retained. Then, to explore the consistent

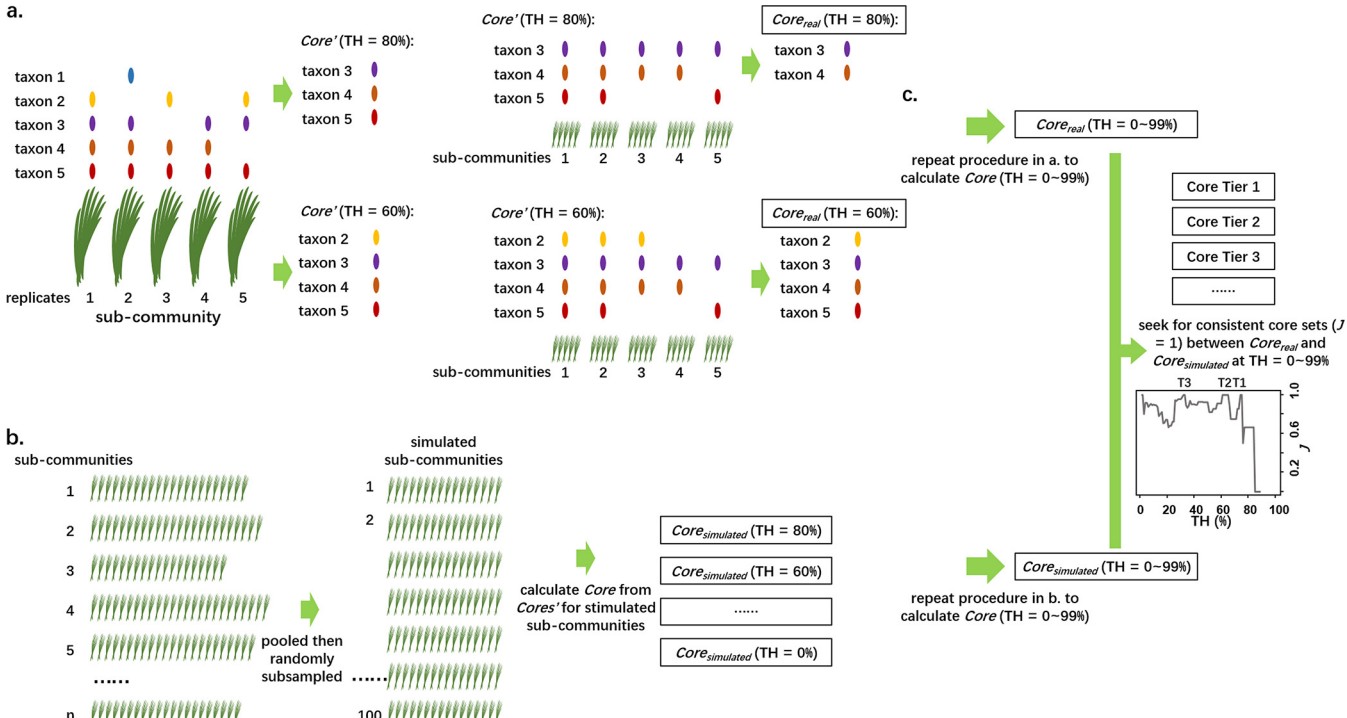

**FIG 5** Conceptualization of the core microbiome used in present study. a) Core set determined at certain threshold based on real sub-community data. The core' set in a given sub-community (or local community) is determined with the frequencies of taxon occurrences. Taxa with frequencies above threshold are regarded as members in core' set of the sub-community, as described in Equation 1. When a taxon is determined as core' member with higher frequency than threshold among all sub-communities, it is determined as a member in core set, as described in Equation 2. For example, taxon 3, 4, and 5 are diagnosed as core' member in a sub-community when threshold = 80%. Nevertheless, the epidemic taxon 5 is denied by core set later, as its "core" pattern is not adequately prevalent. b) Core set determined at certain threshold based on simulated sub-community data. Replicates from all sub-communities are pooled to meta-community, from which random subsample are proceeded to generate stimulated sub-communities. Then the core set at certain threshold is determined with the same procedures in panel a. c) Core microbiome with statistical support. Serial core sets would be produced by repeating the core set determination at different thresholds (from 0~99%) for real or stimulated community data. Commonly there would be several threshold intervals in which the real and stimulated datasets yield same core sets (Jaccard Index between Core_real and Core_stimulated set equals to 1). The core sets determined with thresholds in such intervals were named as Tier 1, 2, 3......N cores. Researchers could select proper tiers as core criteria in researches due to specific application scenarios. TH, threshold; T, tier; $J$, Jaccard Index.

co-occurrence associations among the subcommunities within a meta-community, the sum values of associations among all subcommunities were illustrated in a combined network graph. The degree (numbers of edges), closeness centrality, and betweenness centrality of each node and the frequency of edges in the combined network graph were calculated to infer the core interactions and the network characteristics between the core and noncore partitions in the network (3, 5). In addition, simulated communities were used to calculate the shared co-occurrence associations following the same methods and criteria. In addition, the dissimilarities ($d$) between the co-occurrence network structures were calculated using the methods described by Schieber et al. (107) to evaluate the comparability of networks generated with real or simulated data sets. The $d$ between the networks was computed considering w1 = 0.45, w2 = 0.45, and w3 = 0.1, and the distance matrix was ordinated with NMDS as described above.

**Data availability.** The demultiplexed pyrosequencing data and associated metadata have been deposited in the Sequence Read Archive, National Center for Biotechnology Information (SRA, NCBI [www.ncbi.nlm.nih.gov/sra]), under the BioProject IDs PRJNA592195 and PRJNA967126.

## SUPPLEMENTAL MATERIAL

Supplemental material is available online only.
**SUPPLEMENTAL FILE 1**, DOCX file, 0.9 MB.
**SUPPLEMENTAL FILE 2**, XLSX file, 1.5 MB.
**SUPPLEMENTAL FILE 3**, TXT file, 0.1 MB.

## ACKNOWLEDGMENTS

We appreciate Qi-Ming Wang at Hebei University and Institute for Cereal Crops Research, Tel Aviv University, for their support on computing resources.

This work was supported by Advanced Talents Incubation Program of the Hebei University (521100221030), Introducing Overseas Talents Funding Project of Hebei Province (C20220512), Chief Scientist Israel Ministry of Agriculture grant 383/15, and Special Project for the Incubation of Scientific and Technological Innovation of College and Middle School Students in Hebei Province (22E50038D).

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
