## [Reviewer comments · Microbiology Spectrum]

Microbiology Spectrum

Distinct features based on partitioning of endophytic fungi of cereals and other grasses

Xiang 孙(Sun), Or Sharon, and Amir Sharon

Corresponding Author(s): Xiang 孙(Sun), Hebei University

Review Timeline:

Submission Date:	February 9, 2023
Editorial Decision:	March 10, 2023
Revision Received:	April 12, 2023
Accepted:	April 18, 2023

Editor: Florian Freimoser

Reviewer(s): The reviewers have opted to remain anonymous.

Transaction Report:

DOI: <https://doi.org/10.1128/spectrum.00611-23>

March 10, 2023

Dr. Xiang 孙(Sun)
Hebei University
School of Life Sciences
No. 180 Wusi Dong Road
Lian Chi District
Baoding, Hebei 071002
China

Re: Spectrum00611-23 (Distinct features based on partitioning of the endophytes of cereal and other grasses)

Dear Dr. Xiang 孙(Sun):

Thank you for submitting your manuscript to Microbiology Spectrum. Your article has been reviewed by two experts, who both find the results interesting and valuable, but also suggest corrections and improvements. If you can incorporate these comments, we would be happy to receive a revised version of your manuscript.

Link Not Available

Sincerely,

Florian Freimoser

Journals Department
Reviewer comments:

Reviewer #1 (Comments for the Author):

Sun et al. investigated the endophytic fungi of wheat and five other related plant species. They provided a unique concept of core- and non-core fungal assemblies. Overall the paper adds interesting information to our understanding of fungal endophytes. However, the article still needs considerable improvement.

- The authors focus a lot on testing the core-, non-core fungal assemblies while missing some basic information about the whole

experiment. For example, what plant tissue (leaf? stem?) did the authors sample? How many replications per site/plant? What are the criteria to determine which plant individual to sample? This information should be provided in M&M (ca. Ln121). I later found out this paper sampled "stem" for endophytes, but this was first brought up in Ln318 in the results. This is very important and should be mentioned in the abstract. It was also unclear if the samples were collected for two years or if the samples were collected between 2017 and 2018 (Ln125). If so, the temporal factor will at least be mentioned in the discussion.

-Several figures include multiple subfigures, making the figure and legend very small to read (e.g. Fig. 2, 4). In Fig. 2C, the brown and purple dots are so small that I couldn't identify them. The resolution needs to be improved, too. People indeed pay less attention to the supplementary figures, but Fig. S3 is really unreadable. I suggest the authors to reorganize the figures, and to think about what are the key figures to keep in the main document.

-Sometimes, I'm not sure if wheat refers to only "bread wheat" or "bread wheat & wild wheat". The figures' legends use scientific names, while general names were used a lot in the text, which is also confusing.

-Ln300: Clarify "abundance" and "prevalence"

-Part of the results was rather descriptive. For example, Ln327: "the curves in the five wild species showed clear downward trends...". However, it's very hard to tell from Fig. 4.

Some minor points:

-Title: Suggest to add "fungi", as endophytes include bacteria, too.

-The discussion on Basidiomycetous yeast can be shortened. Yes, it's interesting, but it has also been shown in several previous studies.

-Fig. 5. The scientific names need can be spelled out.

-Ln241: Why mention S genome and AB genome?

-The use of "Ae." peregrina and "A." peregrina should be consistent (e.g. Ln271 & 273)

Reviewer #2 (Comments for the Author):

Sun et al. attempted to investigate the community dynamics behind the Fungal Endophytic Communities (FECs) in natural and cultivated plants. To engage in this investigation, they leveraged the FECs from five natural cereal species and one commercial wheat plant species. They analyzed the community assembly dynamics based on the fungal compositions across samples from different environments. Specifically, they first proposed a way to define the core vs. non-core communities, and such a division is carried to all following analyses. Second, they analyzed the neutral community dynamics based on the fitting of the neutral community model and also used the DOC (Dissimilarity-Overlap Analysis) analysis to decipher the universality of dynamics. Finally, they constructed the occurrence network between microbes. I am quite open to looking at a revised version if the authors could address some major and minor issues in a satisfactory fashion. I believe that the authors may benefit from the following comments:

Major issues:

1. The schematic Figure 1 is not very helpful. First, it does not properly capture the definition of "core microbiome" in various papers. It is very hard for me to see the connection between each panel and each listed paper in the introduction. Second, in panel d, which delaminates the method they proposed here, it is very hard for me to see the meaning of tiers here. I strongly suggest they revise the schematic figure based on the previous and their definitions and try to show the gist of those definitions.
2. Although various past definitions of "core microbiome" were listed, why authors need to introduce a new modified definition is unclear after reading the introduction. It would be great if the authors could discuss some potential issues of past definitions and why it may be important to modify the definition for the endophytes. Have anyone investigated the endophytes using the concept of "core microbiome"? What have they achieved? If so, please summarize them and tell me why a modification is necessary. If not, please motivate readers by explaining that this perspective hasn't been explored in the past and why it fits the scenario of endophytes here.
3. Lines 275-279: The choice of a cutoff threshold for determining the core vs. non-core as close to 50% of total relative abundances sounds a bit too arbitrary for me since there are many other ways to choose. For example, one can simply choose a cutoff threshold as the total fraction of selected core taxa among all taxa close to 5% (or other values). Could the authors demonstrate that their following results are not particularly sensitive to their current choices by testing their results on more feasible definitions (such as the simple I proposed)?

Minor comments:

1. Line 15: FEC was mentioned without the full name being defined. Please show the full name before the acronym occurs.
2. Section of "Quality control and taxonomy assignment": please specify whether amplicon sequencing or whole-genome metagenomic sequencing is used. According to what I understood, amplicon sequencing was used. Is 16S, 18S, or ITS2

amplicon sequencing used? I guess the 18S or ITS2 is more appropriate for fungal communities. Please specify them and the particular genetic sequences that are amplified in the Methods section instead of only briefly mentioning some details in the results.

3. What is the connection between core and non-core species? Are they stratified by different sizes in ecological niches (P Brun et al., *Limnol. Oceanogr.*, 2015)? Or are they linked by cross-feeding interactions and coming from different trophic levels (SA Shetty, et al., *npj Biofilms and Microbiomes*, 2022; T Wang et al., *PloS Computational Biology*, 2019)? Some discussion may help readers to understand the underlying mechanism beyond the data analyses.

4. Figure 3: a text description of shared x-axes is missing in the figure, though it is described in the figure caption. I would suggest they add an x label for all figures. In addition, there is not a y label for the grey lines (i.e., J as the threshold changes)

5. Lines 294-295: what are Nm values? It seems to be only briefly mentioned in the figure caption (lines 343-344). Please briefly introduce its meaning in the text directly so that it is understandable to readers who are not familiar with the neutral theory of species diversity.

6. Figure 4: once again, it would be great if the meaning of colors in the color bars and the difference between two bar plots (upper vs. bottom) could be demonstrated in the figure instead in the figure captions.

7. Figure 5: colors of vertices are barely visible because the figure is overwhelmed by the blue color of edges. Either reduce the edge width of edges or increase the node sizes to highlight the importance of node colors.

Staff Comments:

Preparing Revision Guidelines

Please return the manuscript within 60 days; if you cannot complete the modification within this time period, please contact me. If you do not wish to modify the manuscript and prefer to submit it to another journal, please notify me of your decision immediately so that the manuscript may be formally withdrawn from consideration by Microbiology Spectrum.

All authors deeply appreciated comments of reviewers and editor, and their efforts to improve the manuscript. We have addressed all the comments and modified the manuscript accordingly. We also updated publication information of two bioRxiv papers in the Reference section.

Please see below our detailed response:

Reviewer #2 (Comments for the Author):

Major issues:

1. The schematic Figure 1 is not very helpful. First, it does not properly capture the definition of "core microbiome" in various papers. It is very hard for me to see the connection between each panel and each listed paper in the introduction. Second, in panel d, which delaminates the method they proposed here, it is very hard for me to see the meaning of tiers here. I strongly suggest they revise the schematic figure based on the previous and their definitions and try to show the gist of those definitions.

We have modified Fig. 1 to focus on our definitions. Panels a, b, and c were removed, since their definition have been previously illustrated in original literatures. We produced a new panel d to describe details of our idea.

2. Although various past definitions of "core microbiome" were listed, why authors need to introduce a new modified definition is unclear after reading the introduction. It would be great if the authors could discuss some potential issues of past definitions and why it may be important to modify the definition for the endophytes. Have anyone investigated the endophytes using the concept of "core microbiome"? What have they achieved? If so, please summarize them and tell me why a modification is necessary. If not, please motivate readers by explaining that this perspective hasn't been explored in the past and why it fits the scenario of endophytes here.

Since the paradigm of "core microbiome" has been well accepted, there have been many studies that applied the core concept in endophyte research. Please see several examples in the following. There are also online tools to detect core microbiome for various types of microbial communities, such as MetaCoMET (1), which use membership, composition, and persistence methods as discussed by Shade and Handelsman (2). Chen *et al.* applied MetaCoMET in their research to detect core endophyte of the medicinal plant *Salvia miltiorrhiza* based on membership and persistence methods (3). Some studies developed their own concepts of core, different from mainstream definitions, which we reviewed in the manuscript. Examples are cited below and include early studies such as Lundberg *et al.* (2012) work on the core root microbiome of *Arabidopsis thaliana*, who considered enriched taxa as core endophyte, and Thomas *et al.* (2019) who defined core endophytic mycobionemes of

trees as those showed strong co-occurrence associations with a host. More recently, increasing number of studies adopted the abundance-occupancy distributions to detect cores in endophytic microbial communities (Grady, 2019; Liu 2020)), which is mainly proposed by Shade (2019). There are also studies that used t only he terminology without clarifying the definition and criterion of core such as Kuzniar (2020). We added these examples in the Introduction.

There are several problems in current studies referring to core microbiome: 1) usually, only arbitrary criteria that rely on empirical reasons are used, which is not tested with statistical approaches, 2) arbitrary criteria might decrease the resolution of research results and lead to lose of information, 3) concepts of core based on abundance can be largely biased by epidemic distribution, 4) because of heterogeneity among natural microbial communities, core set reduces sharply as numbers of sub-communities increases, especially when core is defined with overlapping idea.

Core members of microbial communities are expected to play pivotal roles in organizing the dynamics of resident microbiomes, therefore, increasing studies have incorporated core identification to their research strategy. However, core concepts might be misused in some studies, and on the other hand, some researchers oppose the whole concept of core microbiome. Here, we introduce a novel definition and detection method of core members of a microbial community trying to address these inherent problems. We feel that our concept has properly addressed the issue of core taxa in our study, and hope it is applicable in a wide range of studies.

1. Wang Y, Xu L, Gu YQ, Coleman-Derr D. 2016. MetaCoMET: a web platform for discovery and visualization of the core microbiome *Bioinformatics* 32:3469–3470.
2. Shade A, Handelsman J. 2012. Beyond the Venn diagram: the hunt for a core microbiome. *Environmental Microbiology* 14:4-12.
3. Chen H, Wu H, Yan B, Zhao H, Liu F, Zhang H, Sheng Q, Miao F, Liang Z. 2018. Core microbiome of medicinal plant *Salvia miltiorrhiza* seed: A rich reservoir of beneficial microbes for secondary metabolism? *International Journal of Molecular Sciences* 19:672.
4. Lundberg DS, Lebeis SL, Paredes SH, Yourstone S, Gehring J, Malfatti S, Tremblay J, Engelbrekton A, Kunin V, del Rio TG, Edgar RC, Eickhorst T, Ley RE, Hugenholtz P, Tringe SG, Dangl JL. 2012. Defining the core *Arabidopsis thaliana* root microbiome. *Nature* 488:86+.
5. Thomas D, Vandegrift R, Roy BA, Hsieh H-M, Ju Y-M. 2019. Spatial patterns of fungal endophytes in a subtropical montane rainforest of northern Taiwan. *Fungal Ecology* 39:316-327.
6. Grady KL, Sorensen JW, Stopnisek N, Guittar J, Shade A. 2019. Assembly and seasonality of core phyllosphere microbiota on perennial biofuel crops. *Nature Communications* 10:10.
7. Liu D, Howell K. 2020. Community succession of the grapevine fungal microbiome in the

annual growth cycle. Environmental Microbiology doi:10.1111/1462-2920.15172:16.

8. Shade A, Stopnisek N. 2019. Abundance-occupancy distributions to prioritize plant core microbiome membership. *Current Opinion in Microbiology* 49:50-58.
9. Kuzniar A, Wlodarczyk K, Grzadziel J, Goraj W, Galazka A, Wolinska A. 2020. Culture-independent analysis of an endophytic core microbiome in two species of wheat: *Triticum aestivum* L. (cv. 'Hondia') and the first report of microbiota in *Triticum spelta* L. (cv. 'Rokosz'). *Systematic and Applied Microbiology* 43:126025.

3. Lines 275-279: *The choice of a cutoff threshold for determining the core vs. non-core as close to 50% of total relative abundances sounds a bit too arbitrary for me since there are many other ways to choose. For example, one can simply choose a cutoff threshold as the total fraction of selected core taxa among all taxa close to 5% (or other values). Could the authors demonstrate that their following results are not particularly sensitive to their current choices by testing their results on more feasible definitions (such as the simple I proposed)?*

As discussed above, arbitrary criteria have been widely used in microbiome studies. However, we did not use an arbitrary criterion, but rather defined a core when the simulated sub-communities and real local communities met provided the same core sets. According to our approach, there are several tiers of core sets defined by corresponding thresholds. The cutoff of close to 50% of total RA was defined based on the consideration of symmetric data in subsequent statistical analysis. The cutoffs could certainly be determined around 1st tier or other tiers.

Alternatively, we might want to test the result at a different criteria, for example when core taxa account for 5% of the total taxa. Please note that there is no consistent set yielded from stimulated and real sub-communities (where $J = 1$) around this cutoff. In this case, we used only the core set yielded from real local communities, which is still a core set according to our definition but lacks statistical support. Please see Fig. R1 for more information.

Figure response 1. Taxon number and abundance in the core sets of real and simulated communities, modified from Fig. 3. The horizontal ordinates indicate thresholds ranging from 0% to 99%. Colored solid lines show the number of core taxa, and the dashed lines represent the sum of the relative abundance of core taxa. Dark colors show values calculated with real communities, and light colors show values calculated with simulated communities. Gray solid lines indicate J between core sets

identified with real and simulated communities at certain thresholds. Horizontal red solid lines indicate 5% of all taxa. Vertical red dashed lines indicate prevalence thresholds for core sets at “5% of all taxa” criterion.

With the 5% taxa core criterion, 25, 22, 25, 24, 24, and 17 taxa were determined as core members in FECs of *Ae. peregrina*, *Ae. sharonensis*, *A. sterilis*, *H. spontaneum*, *T. aestivum*, and *T. dicoccoides*, respectively (Table R1). Please see file “new cores.xlsx” for the lists of core taxa for the six plant species.

Table response 1. Summary for 5% taxa core criteria.

	No. of all taxa	Overall abundance	core taxa determined	Thresholds (%)	Abundance sums of cores	RA sums of cores
Aegilops peregrina	491	518.2094	25	40	354.0824	0.683281
Aegilops sharonensis	438	428.574	22	31	287.2628	0.670276
Avena sterilis	495	524.84	26	22	380.6693	0.725305
Hordeum spontaneum	483	503.549	24	26	358.5792	0.712104
Triticum aestivum	475	719.8923	24	28	517.3873	0.718701
Triticum dicoccoides	379	369.5694	17	34	231.6732	0.626873

Stack bars in Fig. R2a indicate that there are higher proportions of neutral taxa in non-core sets, which is consistent with our previous results. Fig. R2b doesn’t change with alternate criteria. Thus, it is exactly the same as Fig. 4b in the manuscript. Fig. R3c shows DOCs of meta-communities in the core and non-core partitions. Nevertheless, DOC for non-core partitions could not be calculated in all hosts except *Ae. peregrina*, as new criterion produced over sparse datasets. The DOC of non-core partition in *Ae. peregrina* FEC shows similar trend that upward DOC turns to downward curve. In the manuscript, our criterion produced upward or roughly flat curves for core partitions in all hosts (Fig. 4B), which suggested no universal dynamics in the core partition. However, DOCs in core partitions of all hosts but *T. aestivum* are observed firstly upward then downward, which suggests universal dynamics in some degree. As the 5% taxa core criteria expanded the core sets, DOC results indicated that the new coming taxa brought universal dynamics signals into core datasets, and DOCs was sensitive to detect the signals. Meanwhile, results with 5% taxa core criteria confirmed the lack of universal dynamics in FECs of *T. aestivum*, which showed no universal dynamics signal in Fig. 4c, S1, and R2c.

Therefore, the community traits borne in non-core partitions drift to the core partitions along with the decreasing criteria for core. The results indicate that our criteria applied in manuscript and supported by bootstrapping are rational strategy to depict FEC structures of different hosts, and can reliably provide novel insights in understanding structures of microbial communities.

Figure response 2. Neutral community model (NCM) fit and dissimilarity-overlap curves (DOC) of core and non-core taxa with new 5% criterion. The caption is same with **Fig. 4. a.** NCM fit of each taxon. **b.** Taxon proportions for core and non-core partitions at different thresholds. The stack area charts illustrate the proportions of taxa that fit to or deviate from NCM in the core and non-core sets with an increase in the thresholds of core from 0 to 99% (x-axis). **c.** DOC of meta-communities in the core and non-core partitions of each host. DOCs (red curves) are calculated for core (left) and non-core (right) sets to detect universality in the dynamics of endophyte communities in different partitions. As for non-core sets, only *Ae. peregrina* is calculated.

Minor comments:

1. Line 15: *FEC* was mentioned without the full name being defined. Please show the full name before the acronym occurs.

We added the full name.

2. Section of "Quality control and taxonomy assignment": please specify whether amplicon sequencing or whole-genome metagenomic sequencing is used. According to what I understood, amplicon sequencing was used. Is 16S, 18S, or ITS2 amplicon sequencing used? I guess the 18S or ITS2 is more appropriate for fungal communities. Please specify them and the particular genetic sequences that are amplified in the Methods section instead of only briefly mentioning some details in the results.

We used ITS 1 amplicons in our study. We added the information to Methods and Materials. Please see line 160-163.

3. What is the connection between core and non-core species? Are they stratified by different sizes in ecological niches (*P Brun et al., Limnol. Oceanogr., 2015*)? Or are

they linked by cross-feeding interactions and coming from different trophic levels (SA Shetty, et al., npj Biofilms and Microbiomes, 2022; T Wang et al., PloS Computational Biology, 2019)? Some discussion may help readers to understand the underlying mechanism beyond the data analyses.

Thank you for your comment. The connection between core and non-core species has also been a long-standing question for us. In comparison to mycorrhizal and pathogenic fungi, the details of the endophyte community are highly individualized due to the presence of diverse symbionts and are therefore scenario-dependent. It very different for other systems, such as intestine microbiota or aquatic organisms, due the lack of physical barriers and therefore they might be regarded homogeneous environments. It is challenging to depict the biological interaction forms and processes between members of the endophyte community within plant tissues, and deliver this information to readers. Therefore, this manuscript puts forward further questions (for example, does FECs represent a holistic or fragmented systems) rather than ultimate answers based on our findings.

Referring to possible explanations of connection between core and non-core species; At first, the endophytes were all from small pieces of stems from healthy mature plants. It is currently impractical to identify separate niches for FEC members at this microscopic scale. Thus, they should be considered from same niche.

With regards to the niche effect on species stratification, we further calculated the niche breadths of endophyte taxa in FECs in terms of host species, location, and host population (host species \times location) with Levins' index (MicroNiche::levins.Bn in R). The results indicate that core members have higher niche breadths (Bn) with reliable detections (Below.LOQ = N, please see attached "Levins index.xlsx"). The majority of core members had highest Bns referring to host and host population factors. Core members were also among top taxa with high Bns even tested on location. The results indicated that core members occupied broader niches, compared with other non-core members. However, with adjusted *p* value higher than 0.05, which denied statistical significance. This might be caused by the limited numbers of levels investigated for each factor. Thus, we prefer not to show the results of Levins' index in the manuscript.

Moreover, we should keep in mind that our criterion for core is defined based on prevalence, namely the width of distribution, which naturally diagnose members with broad niches.

Referring to the trophic roles of fungi in communities, one usual method is to infer possible functional traits using function annotation tools like Funguild. However, we didn't apply such approaches in the present study. It would be problematic to over-extrapolate the potential functional traits solely based on a function annotation database, as the genetic backgrounds and physical functions of fungi are more complex

than those of bacteria. Additionally, the resolution at which fungi are predicted into function groups by the database is too low to infer the trophic networks within endophyte communities. We have incorporated related discussion in the manuscript.

In summary, we can't reach a clear conclusion about how the core taxa connected to non-cores, but we have incorporated more discussion i to help readers understand possible underlying mechanism.

4. Figure 3: a text description of shared x-axes is missing in the figure, though it is described in the figure caption. I would suggest they add an x label for all figures. In addition, there is not a y label for the grey lines (i.e., J as the threshold changes)

Thank you for your comments. An x label was added. Indeed, J should use the secondary y axis. Annotation was added.

5. Lines 294-295: what are Nm values? It seems to be only briefly mentioned in the figure caption (lines 343-344). Please briefly introduce its meaning in the text directly so that it is understandable to readers who are not familiar with the neutral theory of species diversity.

Nm is the migration rate. It is used to measure gene flow with the number of migrants coming into a population per generation in population genetics (Whitlock and McCauley, 1999, *Heredity* 82, 117–125). In the context of NCM, higher Nm value indicated that sub-communities were more connected to each other. The single-digit Nm in suggests relatively low migration among sub-communities and lead to more isolated and differentiated sub-communities, which echoed the high intra-host variation shown by Fig 2C. We have added a brief introduction, please see line 362-365.

6. Figure 4: once again, it would be great if the meaning of colors in the color bars and the difference between two bar plots (upper vs. bottom) could be demonstrated in the figure instead in the figure captions.

Figure 4 was imprved.

7. Figure 5: colors of vertices are barely visible because the figure is overwhelmed by the blue color of edges. Either reduce the edge width of edges or increase the node sizes to highlight the importance of node colors.

Sorry for the inconvenience. The taxa in the graph are of low RAs and in small size. we have adjusted the figures and believe that now it looks better than the original version.

All authors deeply appreciated comments of reviewers and editor, and their efforts to improve the manuscript. We have addressed all the comments and modified the manuscript accordingly. We also updated publication information of two bioRxiv papers in the Reference section.

Please see below our detailed response:

Reviewer #1 (Comments for the Author):

- The authors focus a lot on testing the core-, non-core fungal assemblies while missing some basic information about the whole experiment. For example, what plant tissue (leaf? stem?) did the authors sample? How many replications per site/plant? What are the criteria to determine which plant individual to sample? This information should be provided in M&M (ca. Ln121). I later found out this paper sampled "stem" for endophytes, but this was first brought up in Ln318 in the results. This is very important and should be mentioned in the abstract. It was also unclear if the samples were collected for two years or if the samples were collected between 2017 and 2018 (Ln125). If so, the temporal factor will at least be mentioned in the discussion.

We tried to keep the text short and therefore did not include these details. In all the study the sampled plant tissues were stems. Plant samples were collected from healthy mature individuals when they started to form spikes. The sampling size is indicated in table S1, which we have now modified to include sampling time. We have revised the Method and Materials accordingly.

In our study, *Ae. sharonensis*, *T. aestivum*, and *T. dicoccoides* samples were collected in March and April of 2017 and *Ae. peregrina*, *A. sterilis*, and *H. spontaneum* samples were collected in March and April of 2018. We planned the sampling time such that the plants were at a similar growth stage and climate conditions. Therefore, the temporal factor is trivial in our opinion and was not analyzed nor discussed in the paper.

-Several figures include multiple subfigures, making the figure and legend very small to read (e.g. Fig. 2, 4). In Fig. 2C, the brown and purple dots are so small that I couldn't identify them. The resolution needs to be improved, too. People indeed pay less attention to the supplementary figures, but Fig. S3 is really unreadable. I suggest the authors to reorganize the figures, and to think about what are the key figures to keep in the main document.

Thank you for your comments. We have improved the figures as suggested.

-Sometimes, I'm not sure if wheat refers to only "bread wheat" or "bread weat & wild wheat". The figures' legends use scientific names, while general names were used a lot in the text, which is also confusing.

We intended to deliver popular information of the species here. We added the scientific names in the revised manuscript.

-Ln300: Clarify "abundance" and "prevalence"

The "abundance" is the total amount of certain endophyte, which is defined as the sum of Hellinger transformation values of reads of the certain endophyte in all plant individuals. The "prevalence" is occurrence ratio of a certain endophyte among plant individuals, which is defined as the numbers of plant individuals colonized by a certain endophyte divided by the number of all plant individuals. We added the definition in Methods and Materials, lines 186-187 and 194-198.

-Part of the results was rather descriptive. For example, Ln327: "the curves in the five wild species showed clear downward trends...". However, it's very hard to tell from Fig. 4.

Sorry. The trends should be described as first upward then downward.

In Bashan's methods (Bashan et al. 2016, Nature 534:259-262), the change point where the DOC (red curves) turns from upward to downward would be indicated with vertical line as in Fig. 4b. If DOC keeps upward trend which suggested no signal for universal dynamics, the vertical line would be located at the rightmost of x-axis.

In our study, we found interesting results in real data (please keep in mind that the real data here is relative to Null data generated in DOC calculation, comparing to real data sets vs simulated data sets in other parts of the manuscript) of non-core partitions which is illustrated at upper right in each panel. The DOC showed clear trends of first upward then downward, which can be viewed directly in *Ae. peregrina*, and *H. spontaneum*, where the vertical lines were located in the middle or near the middle of x-axis. DOCs in *Ae. sharonensis*, *A. sterilis*, and *T. dicoccoides* seemed to reach plateau or slightly downward zone, and change points were also labelled at the x-axis. However, DOC of *T. aestivum* non-core partition kept going upward, and thus the vertical line was labelled at the rightmost of the x-axis.

We revised the results to interpret figure 4b in a concise way.

Some minor points:

-Title: Suggest to add "fungi", as endophytes include bacteria, too.

Thanks. It has been added.

-The discussion on Basidiomycetous yeast can be shortened. Yes, it's interesting, but it has also been shown in several previous studies.

The EBY section has been shortened.

-Fig. 5. The scientific names need can be spelled out.

Sorry for the inconvenience. They are fully spelled now.

-Ln241: Why mention S genome and AB genome?

Sorry, this is a mistake that was carried over from an earlier version. .

-The use of "Ae." peregrina and "A." peregrina should be consistent (e.g. Ln271 & 273)

Thanks. They were uniformed as "Ae."

Aegilops_peregrina	Aegilops_sharonensis	Avena_sterilis	Hordeum_s
Alternaria_angustiovoidea	Acremonium_F_OTU_746	Alternaria_angustiovoidea	
Alternaria_infectoria	Alternaria_angustiovoidea	Alternaria_infectoria	
Candida_sake	Alternaria_infectoria	Candida_sake	
Cladosporium_F_OTU_962	Candida_sake	Cladosporium_exasperatum	
Cladosporium_F_OTU_965	Cladosporium_exasperatum	Cladosporium_F_OTU_962	
Cladosporium_grevilleae	Cladosporium_F_OTU_962	Cladosporium_F_OTU_963	
Dioszegia_buhagiarii	Cladosporium_F_OTU_965	Cladosporium_F_OTU_965	
Dioszegia_hungarica	Cladosporium_grevilleae	Cladosporium_grevilleae	
Filobasidium_chernovii	Cladosporium_sphaerospermum	Cladosporium_halotolerans	
Filobasidium_F_OTU_329	Filobasidium_chernovii	Cladosporium_sphaerospermum	
Filobasidium_F_OTU_332	Filobasidium_F_OTU_329	Cystobasidium_benthicum	
Filobasidium_oeirensis	Filobasidium_magnum	Cystobasidium_F_OTU_178	
Mycosphaerella_tassiana	Filobasidium_oeirensis	Debaryomyces_F_OTU_600	
Papiliotrema_frias	Mycosphaerella_tassiana	Filobasidium_chernovii	
Parastagonospora_nodorum	Parastagonospora_phoenicicola	Filobasidium_F_OTU_329	
Phaeosphaeria_F_OTU_624	Phaeosphaeriaceae_F_OTU_675	Filobasidium_magnum	
Pyrenophora_japonica	Sarocladium_implicatum	Filobasidium_oeirensis	
Rhodotorula_babjevae	Sporobolomyces_roseus	Mycosphaerella_tassiana	
Septoriella_hirta	Stemphylium_majusculum	Naganishia_globosa	
Sporobolomyces_roseus	Vishniacozyma_carnescens	Phaeosphaeria_F_OTU_624	
Symmetrospora_coprosmae	Vishniacozyma_heimaeyensis	Pyrenophora_japonica	
Vishniacozyma_carnescens	Vishniacozyma_victoriae	Rhodotorula_babjevae	
Vishniacozyma_dimennae		Sporobolomyces_roseus	
Vishniacozyma_heimaeyensis		Vishniacozyma_carnescens	
Vishniacozyma_victoriae		Vishniacozyma_heimaeyensis	
		Vishniacozyma_victoriae	

spontaneum	Triticum_aestivum	Triticum_dicoccoides
Alternaria_angustiovoidea	Acremonium_F_OTU_746	Acremonium_F_OTU_746
Alternaria_infectoria	Alternaria_angustiovoidea	Alternaria_angustiovoidea
Blumeria_graminis	Alternaria_infectoria	Alternaria_infectoria
Candida_sake	Candida_sake	Candida_sake
Cladosporium_F_OTU_962	Cladosporium_F_OTU_962	Cladosporium_F_OTU_962
Cladosporium_F_OTU_965	Cladosporium_F_OTU_963	Cladosporium_F_OTU_965
Cladosporium_grevilleae	Cladosporium_F_OTU_965	Cladosporium_grevilleae
Cladosporium_halotolerans	Cladosporium_grevilleae	Cladosporium_sphaerospermum
Cladosporium_sphaerospermum	Cladosporium_sphaerospermum	Filobasidium_chernovii
Cystobasidium_F_OTU_178	Cystobasidium_sloffiae	Filobasidium_F_OTU_329
Debaryomyces_F_OTU_600	Cystofilobasidium_macerans	Filobasidium_magnum
Filobasidium_chernovii	Debaryomyces_F_OTU_600	Filobasidium_oeirensis
Filobasidium_F_OTU_329	Filobasidium_chernovii	Mycosphaerella_tassiana
Filobasidium_magnum	Filobasidium_magnum	Sarocladium_implicatum
Filobasidium_oeirensis	Filobasidium_oeirensis	Sporobolomyces_roseus
Mycosphaerella_tassiana	Mycosphaerella_tassiana	Vishniacozyma_carnescens
Naganishia_globosa	Parastagonospora_nodorum	Vishniacozyma_heimaeyensis
Papiliotrema_frias	Parastagonospora_phoenicicola	
Pyrenophora_japonica	Rhodotorula_babjevae	
Rhodotorula_babjevae	Sarocladium_implicatum	
Sporobolomyces_roseus	Sporobolomyces_roseus	
Vishniacozyma_carnescens	Vishniacozyma_carnescens	
Vishniacozyma_heimaeyensis	Vishniacozyma_heimaeyensis	
Vishniacozyma_victoriae	Vishniacozyma_victoriae	

red

adj p > 0.05

bold and yellow highlight

core

April 18, 2023

Dr. Xiang 孙(Sun)
Hebei University
School of Life Sciences
No. 180 Wusi Dong Road
Lian Chi District
Baoding, Hebei 071002
China

Re: Spectrum00611-23R1 (Distinct features based on partitioning of endophytic fungi of cereals and other grasses)

Dear Dr. Xiang 孙(Sun):

Your manuscript has been accepted, and I am forwarding it to the ASM Journals Department for publication. You will be notified when your proofs are ready to be viewed.

Sincerely,

Florian Freimoser
Editor, Microbiology Spectrum
